# N-Acetylated Monosaccharides and Derived Glycan Structures Occurring in N- and O-Glycans During Prostate Cancer Development

**DOI:** 10.3390/cancers16223786

**Published:** 2024-11-10

**Authors:** Tomas Bertok, Eduard Jane, Michal Hires, Jan Tkac

**Affiliations:** 1Institute of Chemistry, Slovak Academy of Sciences, Dubravska cesta 9, 845 38 Bratislava, Slovakia; 2Glycanostics, Kudlakova 7, 841 08 Bratislava, Slovakia

**Keywords:** glycosylation, glycan, lectin, prostate cancer, PSA, LacdiNAc

## Abstract

Prostate cancer is globally one of the commonest cancer types with hundreds of thousands of deaths annually. Early diagnostics and the elimination of overtreatment lead to higher survival rates and increase the overall quality of life. Novel diagnostic markers are continually being introduced, but glycans, as one of the most promising cancer-related biomarkers, are excluded due to technological requirements, despite the fact that several approaches compatible with clinical practice have emerged in recent years.

## 1. Introduction

Early in the 20th century, it became apparent that sugar molecules are central to cell metabolism and are closely connected to the metabolism of other biomolecules. Notable scientists and Nobel laureates such as Emil Fischer, Paul Ehrlich, Otto Warburg, Hans Krebs and Andrew Benson were all focused on their concept of glycolysis, respiration/sugar “combustion” and hydrogen transfer using thin slices of tissues and manometers [1,2]. The findings of this era led to the discovery of the so-called Warburg effect—a shift in cancer metabolism to growth promotion by the formation of lactate even in the presence of oxygen and functional mitochondria [3]. The fact of cancer cells’ metabolism and signaling being in some specific pathways different from healthy cells (being unaffected) has subsequently been thoroughly examined, although the major focus has been directed towards the DNA changes affecting the cell “behaviour”, i.e., mutations in a genetic code. Gabius et al. often refer to glycans (complex and branched saccharides) as a “sugar code” [4]. Although the structure of glycans cannot be directly predicted from the genetic code (unlike for proteins), changes in this glycocode reflect the physiological state of an organism in real time, as compared with DNA mutations which do not necessarily mean the presence of a disease. Furthermore, the main evolutionary advantage is that changes in the glycocode affecting many physiological processes do not have to be associated with changes in the genetic material, which needs to be preserved, rendering glycans a useful tool for clinical diagnostics [5,6]. In vivo glycoconjugates are the result of a post-translational enzyme-driven process denoted as glycosylation, taking place in the endoplasmic reticulum and the Golgi apparatus. Glycosylation involves the addition of N-linked glycans (attached to the ASN-X-SER/THR sequence, where X ≠ PRO), O-linked glycans (attached to SER/THR), phosphorylated glycans, glycosaminoglycans and glycosylphosphatidylinositol (GPI) anchors to peptide backbones and mannose to tryptophan (C-mannosylation) [7].

Prostate cancer (PCa) is one of the most common cancers in men worldwide, where early diagnosis and treatment can be lifesaving. Glycosylation, as a most common post-translational modification of proteins (including the commonly used PSA biomarker, among others), was shown to be altered and correlated with the level of aggressiveness and stage of the disease [8,9]. Although well-regulated in non-pathological states, it mirrors the importance of glycans in cell–cell interactions, communication and adhesion. Among other structures present in glycoconjugates in vivo, GalNAc is of great importance in cancer research, as it is a crucial part of N-glycans and also truncated O-glycans during cancerogenesis. For instance, the MUC1 glycoprotein is overexpressed in many cancer tissues, carrying cancer-related Tn and sTn antigens (GalNAc-α-1-O-THR/SER and Neu5Acα2-6GalNAcα1-O-SER/THR, respectively) [10]. In a previous paper, we discussed the importance of sialic acid (N-acetylneuraminic acid, Neu5Ac) in different types of cancers [11]. In the present study, we focus on the biochemistry of formation, recognition (by antibodies, lectins and enzymes such as GalNAc- and other glycosyltransferases, GT) and analysis of the GalNAc-containing glycan structures related to prostate cancer development and progression, as these changes may initiate or promote epithelial-to-mesenchymal transition and metastases’ formation.

## 2. Bioprocesses Behind and Implications for GalNAc-Containing Glycans

Heavily O-glycosylated membrane-bound or extracellular proteins are denoted as mucins. The mucin molecules carry one of the eight different core structures derived from the attachment of UDP-GalNAc to the SER/THR of the protein backbone by ppGalNAcTI (the so-called Tn antigen). Other structures emerge as this GalNAc is further modified by other enzymes, such as ST6GalNAcI or T synthase. These modifications lead to the formation of the sialyl Lewis antigens, sLex and sLea [12]. As early as 1975, the so-called Thomsen–Friedenreich T antigen (Galβ1-3GalNAcα1-Ser/Thr disaccharide) was shown to be heavily expressed in breast cancer tissues in a pioneering work by Springer et al. [13]. The analysis of these structures remains challenging using affinity-based methods (involving glycan-recognizing molecules), because they may not be readily available for biorecognition, as opposed to more accessible N-glycans. Mucin structures, more specifically GalNAc modifications, are controlled by 20 GalNAc transferase isoenzymes (GalNAcTs) and are often referred to as canonical gel-forming (MUC2, 5AC, 5B, 6 and 19), canonical non-gel forming (MUC7 and 9), canonical membrane-bound (MUC1, 3, 4, 12, 13, 14, 15, 16, 17, 18, 20, 21 and 22) and non-canonical structures bound to well-known proteins such as CD43 [14]. These mucin domains have been proposed as diagnostic markers and also therapeutic targets not only for prostate, but also for other types of cancers. Secreted and transmembrane mucins have some features in common, such as the presence of von Willebrand D (VWD) and a highly glycosylated core containing repeating sequences rich in Pro, Ser and Thr, the so-called PTS domain. These glycans are present at the surfaces of the respiratory, gastrointestinal and genito-urinary tracts, where they create a protective barrier for epithelial cells against infection, dehydration and possible damage [15]. Recently, an engineered bacterial mucin-selective protease fused to a cancer antigen-binding nanobody has been shown to form a conjugate selectively degrading cancer cells mucins; this promoted cell death in culture models and reduced tumor growth in mice [16]. Monosaccharide units necessary for the biosynthesis of glycans are often synthesized from glucose, which is imported into cancer cells in higher levels due to an increased expression of glucose transporters on the cell membrane (GLUT), as cancer cells switch their metabolism to biomass synthesis rather than the aerobic oxidation of glucose to generate energy (36 molecules of ATP per single glucose molecule) [17]. A schematic presentation of the different metabolic pathways and enzymes involved in this process is depicted in Figure 1 [18].

In the literature, more attention is currently paid to the N-glycosylation of prostate cancer biomarkers (or possible biomarkers) than to O-glycosylation [19]. There are, however, several potential O-glycosylation sites in the molecule of PSA (also PAP and MUC1)-bearing sLex glycans as confirmed by ELISA and RT-PCR of the transcript levels of GCNT1 transferase in prostate cancer cells [20]. N- and O-glycan structures bearing Gal and GalNAc residues (such as poly-LacNAc or LacdiNAc, which can be associated with sensitivity to chemotherapy) [21] recognized and recognizable by naturally occurring lectin molecules, in this case, the rHEL3 isolectins from *Hericium erinaceus* mushroom and WFL from the *Wisteria floribunda* plant, respectively [22,23], could be effectively used in routine clinical diagnostics since it does not depend on costly and time-consuming mass spectrometry. Combination with other clinical characteristics, such as PSA velocity or PSA density, would probably further increase the overall diagnostic accuracy. Finally, not only the quality but also the quantity of the glycan molecules can be altered during cancer progression. Among single nucleotide polymorphism in kallikrein family genes, rs61752561 in KLK3 results in a missense mutation and a conversion of Asp102 to Asn102 in PSA, thereby creating a new glycosylation site [24].

A subtype of prostate cancer, so-called mucinous (or colloid) tumors, prostatic adenocarcinoma with mucinous features (i.e., a mucin-producing urothelial-type adenocarcinoma of the prostate), are defined as tumors composed of at least 25% of a mucinous extraluminal component [25]. In 2005, the International Society of Urological Pathology Consensus Conference on the Gleason Grading of prostatic carcinoma suggested grading these tumors as Gleason score 8 (4 + 4) [26], as these tumors—although relatively uncommon—were believed to be a higher risk finding, significantly affecting therapy decisions. This was recently shown not to be true when compared with acinar adenocarcinoma [27]. The diagnostic potential of mucins is attributed to their deregulated expression and aberrant glycosylation, as well as their ability to induce the production of autoantibodies [28]. Among other mucins, such as MUC16 (CEA125) commonly used for ovarian cancer, MUC1 has been investigated and overexpressed in castration-resistant and neuroendocrine PCa, suppressing androgen receptor-signaling and inducing a neural BRN2 transcription factor. In addition, MUC1-C activates an MYC→BRN2 pathway and induces MYCN, EZH2 and NE differentiation markers associated with PCa progression [29]. MUC1 has also been shown to be an independent prognostic biomarker for PCa-associated death and tumor proliferation (for MUC1-positive DU145 and PC3 prostate cancer cell lines) [30,31]. Moreover, genomic alterations in the MUC1 network have also been shown to correlate with PCa relapse [32]. Not only changes in MUC1 levels, but also glycosylation patterns can carry important information about the state of the producing cells. For O-glycosylation in prostate cancer, GCNT1 (sometimes C2GNT1) glycosyltransferase plays an essential role in the formation of core 2 branched O-glycans, promoting chain-branching and elongation. Moreover, expression changes (at gene and protein levels) in GCNT1 have been linked to PCa aggressiveness, disease spread and post-recurrence surgery [33,34]. In the following sections, the focus will be on N-glycosylation, as this is the area already exploited by major players in the pharmaceutical industry and these structures can be readily analyzed using common clinical table-top techniques.

## 3. Aberrant Glycosylation and Its Analysis

Various immune cells interact with their surroundings via cell surface molecules, where glycosylation plays a crucial role in regulating the stability, degradation/half-life and flexibility of these glycoconjugates’ motifs [35,36]. Aberrant glycosylation lead to various signaling pathways being affected, causing conditions such as autoimmune diseases (systemic lupus erythematosus, rheumatoid arthritis and inflammatory bowel disease) or cancer, exhibiting an oncofetal phenotype [7,37]. Because the composition of plasma protein glycans (e.g., autoantibodies affecting B cell activation) affects the above conditions, glycosylation-based interventions currently attract much interest, creating space for glyco-engineered therapeutic proteins. The removal of core-fucosylation is known to enhance antibody-dependent cellular cytotoxicity (ADCC, the presence of this fucose on the conserved N-glycan in the Fc domain decreases its ability to bind via the FcγRIIIa receptor to white blood cells, thus decreasing their ability to kill cancer cells) [38,39]. Hence, the next generation of therapeutic agents are recombinant non-fucosylated/afucosylated antibodies [40,41]. N-glycan structures are generally more conserved than O-glycans, consisting of (i) a trimannosyl core and two GlcNAc residues—this structure is derived from a precursor oligosaccharide, which is processed in the endoplasmic reticulum; (ii) several antennae and/or (bisecting) GlcNAc residue; and (iii) linear oligosaccharide chains of different compositions, usually terminated with sialic acid [42]. The composition is usually dependent on a number of factors, such as the availability of substrates/catalysts, as the Golgi apparatus is fragmented during cancer progression leading to protein dysregulation, or GT gene expression [43,44,45].

Glycosylation as the most-common post-translational modification already starts in the endoplasmic reticulum (ER) by an “en bloc” attachment of a glycan precursor Glc3Man9GlcNAc2 to the protein backbone and is subsequently trimmed and modified by the action of Golgi-localized enzymes. During oncogenesis, harsh conditions and a microenvironment characterized by high metabolic demand and subsequent nutrient limitations caused also by poor vascularization, acidosis and hypoxia disrupt homeostasis in multiple cell types, including calcium ions, leading to the accumulation of misfolded/unfolded proteins even in infiltrating immune cells and cancer cells. As a response to the ER stress, unfolded protein response (UPR) is being activated to restore ER homeostasis [46,47]. Three highly conserved signaling arms exist—IRE1α. PERK and ATF6, playing an important role in tumor growth, invasion, metastasis and resistance to therapy [48]. Targeting the ER stress may thus be an effective strategy to attenuate the protective effect of the stress on cancer cells [49]. UPR has been linked to the androgen receptor signaling pathways—overexpressing BiP chaperones translocating to the cell membrane to elicit cytoprotective effects in PCa, activating the IRE1α pathway via androgen receptor (AR) signaling which in a positive-feedback loop leads to an increased AR expression and downregulating PERK to decrease CHOP-mediated apoptosis [48]. Targeting the IRE1α has been very recently shown to reprogram the microenvironment, thus leading to an enhanced immune response [50]. Interestingly, shaping the immunosuppressive tumor microenvironment by modulating resident tumor-associated macrophages has not yet been fully understood, but has been recently linked to the ER stress-related exosomes in PCa cells, promoting macrophage infiltration and polarization by upregulating the expression of PD-L1, inflammatory factors and the PI3K/AKT pathway [51]. In advanced androgen-refractory PCa, Golgi as the place of N-glycan epitope modifications has been shown to be fragmented and its disassembly correlated with the Gleason score and metastatic status as well as with ST3GAL1 enzyme mislocalization, suggesting several other enzymes in the cascade may be correlated with the disease progression and thus opening a new research area in liquid biopsy approaches and tumor staging [45,52].

These gene signatures might even correlate with cancer subtypes and survivability, such as in the case of HER-2-enriched breast cancers [53]. A schematic presentation of possible cancer-associated N-glycans is depicted in Figure 2. Common changes known to be associated with oncogenesis are increased sialylation and the occurrence of α2,3- and α2,8-bonds, increased branching or fucosylation or occurrence of LacdiNAc (GalNAcβ1-4GlcNAc) and other HexNAc residues, such as bisecting GlcNAc [54,55,56].

Glycosylation plays a crucial role in a multi-faceted process denoted as the epithelial-to-mesenchymal transition (EMT). This process, characterized by changes in the amount of some mRNA transcripts and proteins (such as an increase in mesenchymal N-cadherin and a decrease in epithelial E-cadherin) and also changes in the cell’s shape due to cytoskeleton reorganization, is vital for development and wound-healing [57]. In TGF-β-induced EMT, decreased sialylation, particularly on β4-integrins, has been observed, although sialyltransferases are usually overexpressed (with the exception of ST6GALNAC1) [58]. Another important aspect of EMT glycobiology is the branching of N-glycans by the action of MGAT5 and MGAT3 transferases, which mutually exclude each other’s activity. Increased β1,6-branching leads to metastases (catalyzed by MGAT5, the expression of which is correlated with poor survival), which is inhibited by the occurrence of bisecting GlcNAc in the structure (catalyzed by MGAT3) [59].

It has been shown very recently by Hodgson et al. that sialyltransferase ST6GAL1 (leading to α2-6 sialylated N-glycans) is upregulated in patients with PCa that has spread to the bones, thus possibly promoting bone metastases in vivo and sialylation inhibition blocks the spread of aggressive prostate tumors to the bone [60]. Out of different PCa cell lines, PC-3 seemed to be an aggressive one followed by LNCaP and LuCaP 23.1, showing an upregulation of β1,6-N-acetylglucosaminyltransferase-5b (MGAT5b) and its products β(1,6)-branched oligosaccharides. The same is true for the DU-145 cell line. The binding of PHA-L might be effectively used to distinguish the PCa patients with high accuracy (86.5%), while the signal intensity correlates well with the serum PSA [61]. MGAT5 (sometimes GnT-V) is well known to be responsible for glycan branching in Golgi and to be involved in cancer malignancy [62]. The activity of the enzyme is selectively upregulated by changing cellular N-glycome to immature forms—loss of terminal epitopes leads to an increased amount of the glycan branches, thus involving SPPL3 protease level of MGAT5 cleavage and extracellular secretion is regulated by its own glycan structures [62]. MGAT5 is a glycoprotein that also stabilizes matriptase [63]—protease which suppresses PCa cell invasion by shifting the action of TGF-β towards repressing the EMT [64]. Two different mouse model strains were used, the female C57BL/6 mice from Charles River Laboratories and the female NSG mice from Jackson Laboratory [59]. Epithelial-to-mesenchymal transition (EMT) is basically a tumor cell reactivation and transcriptionally regulated program of the expression of mesenchymal markers such as the above-mentioned N-cadherin and/or other mesenchymal traits and the downregulation of E-cadherin or Zona occludens protein (ZO-1) expression, as well as some metabolic changes including glucose uptake [65,66]. In the transformation process, cells gain the ability to extravasate into the bloodstream. As shown in A459 cells, increased availability of UDP-GlcNAc donor and the subsequent occurrence of aberrant glycoforms is a consequence of a shift toward hexosamine biosynthetic pathway which normally consumes about 2–5% of the total glucose. Also, an increase in α2,6-sialylation, poly-LacNAc and fucosylation has been demonstrated to modulate cancer cell plasticity [58,67]. Interestingly, it has been shown that PD-L1 expression in cancer stem-like cells not only contributes to immune evasion, but EMT transcriptionally induces N-glycosyltransferase STT3 through β-catenin and thus PD-L1 is also stabilized through the regulation of glycosylation [68]. An important player in the EMT is transforming growth factor-beta (TGF-β), also essential in early embryonic development [69]. TGF-β enhances the expression of N-cadherin and fibronectin 1, being also regulated by glycosylation of its signaling pathway components. Interestingly, although hyper-sialylation promotes the metastatic potential of cancer (repulsion leads to detachment of the cells from the primary tumor) cells and thus could be a therapeutic target in oncology [70], sialylation inhibitors may promote EMT and thus have a double-edged effect depending on the cancer cell stage, according to Du et al. [71].

Sialic acid as a part of glycoconjugates usually forms the outermost layer of the cell’s surface. Due to the overexpression of sialyltransferases (STs) during cancer progression, hyper-sialylation occurs in 40–60% of tumors [72]. Hyper-sialylation plays a vital role in cancer development, namely transformation, growth, metastasis and immune evasion [73]. Especially in prostate cancer, the dominant subterminal structure, N-acetyllactosamine (Galβ1-4GlcNAc, LacNAc), is shown to be switched to LDN (GalNAcβ1-4GlcNAc, N,N-diacetyllactosamine or LacdiNAc) by the action of two β4-N-acetylgalactosaminyltransferase (B4GALNT3 or B4GALNT4) enzymes [55]. Interestingly, bisecting GlcNAc in complex N-glycan structures inhibits the action of LDN synthesis by B4GALNT3, thus these structures (recognized by different lectins) are potential prostate cancer biomarkers since LDN affects the malignant properties of a tumor, although the mechanism is not yet fully understood. Moreover, it was recently shown that unlike LacNAc, LDN inhibits the action of some glycosyltransferases, thereby opposing the sialylation and fucosylation of N-glycan antennae [74].

Glycosylation is a highly complex and non-template-driven process with ~300 enzymes involved in the biosynthesis. It is a sequence of transfer and hydrolytic reactions of adding (activated) nucleotide sugar molecules or their removal, reflecting the actual state of a cell; this is one of the main differences with mutations in certain genes, which often only pose a certain risk of developing a condition. FDA-approved oncomarkers are often glycoproteins present in human serum, so glycan analysis is a promising tool for more accurate diagnostics [75,76]. Because glycosylation is a (micro- and macro-) heterogenous process affecting the physico-chemical properties of glycoconjugates (such as their solubility and stability/half-life), robust sample preparation and analytical methods as well as data analyses are required. Advanced tandem mass spectrometry methods (MS/MS) in conjunction with liquid chromatography (LC) or other methods are a gold standard in modern glycoproteomics [77]. Oligosaccharides are the most complex biopolymers due to their ability to create branched structures, variable compositions and linkages, and also due to the presence of isomeric structures and anomers and ion-suppression effects. This is the reason for a sample pretreatment prior to LC, such as graphitized carbon, reverse phase, ion exchange or HILIC chromatography [78]. Glycans can be analyzed in situ (without releasing the glycan) using affinity methods often involving lectins, or they can be released, labelled, separated, chemically modified (e.g., tagged using 2-aminopyridine for isomers separation) [79] and then detected. In both cases, MS/MS is used as an orthogonal method for the validation of structural information [80]. For the release of N-glycans, an enzyme Peptide-N-Glycosidase F (PNGase F) is commonly used. This is an amidase cleaving the innermost GlcNAc from an Asp residue. In this way, almost all the N-glycans can be released, even from a complex sample. However, due to the higher heterogeneity of O-glycans (due to lack of a common core structure), it has been necessary to develop several methods. The enzymatic method using endo-α-N-acetylgalactosaminidase (or sometimes O-glycosidase) can be used in a very limited way, in comparison with the chemical β-elimination method under alkaline conditions producing free glycan and unsaturated peptide [81]. Native glycan analysis is rather challenging, mainly due to the fact that false glycans may occur in cases where the terminal sialic acid is lost during sample preparation, but also because of their low ionization efficiency [82]. For sialic acid stabilization, permethylation (linkage-specific sialic acid permethylation, SSAP) or modifications on a solid phase are performed [83,84]. For the analysis of the presence of LacNAc/LDN structures, less “invasive” methods can be used despite some of the structural information being omitted. These methods involve the use of specific carbohydrate-binding molecules, such as specific antibodies or, more commonly, lectins and their derivatives. Different complex sample glycan analysis approaches are depicted in Figure 3 below.

Lectins are (glyco)proteins of non-immune origin with one or more subunits and one or more carbohydrate-binding sites. Since carbohydrates are structurally very similar, lectins are often described as being quite “promiscuous” as they bind to structurally similar epitopes. Common examples are lectins from the Leguminosae family, such as Con A (ex *Canavalia ensiformis* specific to humans). They can be found in many different species and some may exhibit cytotoxic effects [85]. Artificial molecules mimicking the action of naturally occurring lectins involve boronolectins (based on boronic acid derivatives binding to vicinal diols, often lacking specificity) [86] or bio-engineered neolectins, such as the recently described GalfNeoLect for the recognition of pathogenic microorganism-derived single monosaccharide galactofuranose [87]. The preparation of neolectins is one of the few direct medical applications of computational protein design. An example of this was globotriose(Galα(1,4)Galβ(1,4)Glc)-recognizing Mitsuba-1 protein on the surface of cancer cells with no cytotoxic effects [88]. In the following section, we focus on the natural and commercially available lectins successfully used for PCa diagnostics specific to Gal/GalNAc and GlcNAc.

**Figure 3 cancers-16-03786-f003:**
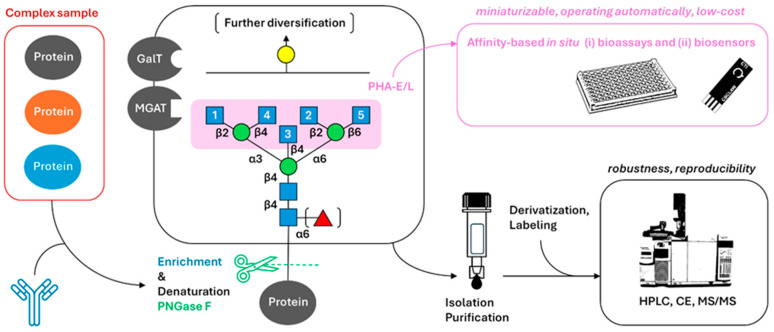
Schematic presentation of different sample analysis algorithms and methodology commonly used in in vitro diagnostic approaches—from left to right. Complex sample comprising of different glycoconjugates could be enriched in specific analytes using highly specific antibodies (red frame on the left) and subsequently the glycan can be released using enzyme PNGase F (in the case of N-glycans). This is common for methods such as HPLC, CE, MS and their combination. In the case of lectin-based methods (ELISA-like bioassays or different biosensors, purple frame—upper right), glycans are often analyzed in situ, without their prior release in a sandwich configuration with antibodies. In the case of different instrumental methods, to ensure the necessary robustness and reproducibility/reliability, chemical derivatization, labelling and/or purification need to be performed (such as sialic acid permethylation—black frame—lower right) [89]. Addition of GlcNAc residues bound via different linkages are catalyzed by MGAT enzyme family members—numbers indicated inside the blue squares; PHA-E/L = *Phaseolus vulgaris* erythroagglutinin/leucoagglutinin [90,91].

## 4. Prostate Cancer—Statistics, Diagnostic Routine and Novel Biomarker Analyses

PCa is the second- to third-most common cancer type diagnosed in men and the fifth leading cause of death, with an annual incidence of ~1.2 M new cases worldwide. Often asymptomatic and/or in an indolent state, it may require only active surveillance in order to prevent overdiagnosis and overtreatment [92]. In 2020, 1,414,259 of new PCa cases (with a cumulative lifetime risk of 3.9%) [93] and 375,304 PCa-related deaths were reported, while the age-standardized incidence rate (very high in selected North European countries) and the age-standardized mortality rate correlate with the human development index positively and negatively, respectively [94,95], although ethnical disparities (highly multifactorial, including socio-economic and biological/genetic factors) are well-known from the US, where African-Americans may have up to a two-fold increased risk for PCa-related death [96,97].

Usually, men after 45 to 50 years of age are advised to undergo a routine prostate cancer screening, increasing the odds of PCa development at this age, although the disease usually has a peak after the age of 70 [98,99]. The screening usually involves digital rectal examination (DRE) and PSA test; diagnosis is based on advanced imaging methods such as (mp)MRI and further laboratory tests, such as fPSA%, PSA density and velocity and/or PHI/4k score kallikrein tests [100]. Historically, prostatic acid phosphatase (PAP) was the first biomarker shown to occur in the plasma of patients with metastatic prostate carcinoma as early as 1938 [101]. The currently used prostate-specific antigen (PSA, kallikrein 3 or KLK3) was later discovered by Flocks and Ablin in the 1960s and 1970s [102]. This is a glycoprotein of ~30 kDa in its free form, which in plasma forms a complex with serine protease inhibitors α−1-antichymotrypsin and α−2-macroglobulin. The complex with α−1-antichymotrypsin (~100 kDa, similar to PAP) is easily measured by common ELISA, unlike the complex with α−2-macroglobulin. The latter complex, however, although unable to hydrolyze large protein molecules, retains some activity towards hydrolyzing small peptide substrates [103,104]. Finally, the last clinically important member of prostate cancer biomarkers is the prostate-specific membrane antigen (PSMA, N-acetylated-α-linked acidic dipeptidase I or glutamate carboxypeptidase II), which is a membrane-bound glycoprotein heavily overexpressed in PCa, thus being extensively studied and used in PCa theranostics. The 177Lu-PSMA-617 therapy has shown promising results in the therapy of PSMA-positive castration-resistant PCa while PSMA-targeting ligands (such as 8F-PSMA-1007 or 68Ga-PSMA-617) are being studied for PCa imaging using PET/CT [105,106]. These three molecules are depicted in Figure 4, including their EC number and molecular mass.

In general, PSA is the first blood test to be performed during a prostate exam, but is not (and should not be) used as a diagnostic tool. To prevent unnecessary prostate biopsies and thus overdiagnosis and overtreatment, it represents a starting line for further testing [107]. These involve other (mostly kallikrein-based) blood tests, such as PHI/PHI density (PHID) or 4k Score panel [108] or imaging using ultrasound (US) or (mp)MRI in the case of ambiguous results. tPSA (total PSA) value in a ratio-to-prostate volume, denoted as PSA density (PSAd), might more accurately indicate the need for a prostate biopsy (AUC value from ROC curve 0.64 vs. 0.78, for PSA and PSAd, respectively) with a sensitivity and specificity of 70% and 79%, respectively [109] and up to 90%+ [110]. A change in the tPSA level over time, the so-called PSA velocity, is involved in active surveillance of men with low grade and indolent PCA for the prediction of the disease progression or as a predictive biomarker in post-treated men [111].

In recent years, many biomarkers and biomarker types have been proposed and introduced for PCa detection; however, the best performance has always resulted from their combination, as in the case of PHID, outperforming PHI and PSAD in the detection of clinically significant PCa (AUC 0.851 vs. 0.819 and 0.813, respectively) [112] or the Stockholm-3 test consisting of a combination of blood biomarkers (fPSA, tPSA, intact PSA, microseminoprotein-β, MIC1, KLK2, genetic polymorphisms (232 SNPs in total) and clinical details such as age, previous biopsies and prostate exams [113]. The possibility to detect and even localize resectable solid tumors using a multi-analyte blood test (CancerSEEK) was demonstrated in 2018 by Cohen et al. [114]; this inspired the search for a combination of different marker types to increase the overall diagnostic accuracy, taking advantage of the tissue- and cancer-specificity of different molecules, including plasma proteins, miRNAs, ctDNA, fusion genes, extracellular vesicles or circulating tumor cells [115,116]. Still largely excluded because of technological obstacles, albeit very accurate, are the post-translational modifications of proteins, mainly glycosylation, as a cancer-specific fingerprint.

## 5. Lac(di)NAc in Prostate Cancer Biomarkers

The regulation of the prostate epithelium and fluid homeostasis is performed by different biomolecules, including kallikreins (PSA and human kallikrein-related peptidase 2, KLK2). Others include Zn^2+^ ions stored in the epithelial cells and causing increased levels of citrate, while the growth and development of the epithelium, including secretory functions, are regulated by androgen steroid hormones mediated by androgen receptor (AR). Suppression of the Krebs cycle (and energy production) and the release of fluid from the gland by accumulation of Zn^2+^ and citrate are caused by 5α-dihydrotestosterone (primary ligand for AR) synthesized from testosterone by the action of 5α-reductase [117,118]. The deregulation of glycosylation pathways of glycoproteins involved in the normal prostate epithelium function was described by Hakomori and later revisited by Kannagi et al.; it is defined as two major concepts—an incomplete synthesis and a neo-synthesis [119,120,121]. From PSA, immunoprecipitated and isolated by using lectin affinity chromatography and later digested and labelled, glycoforms containing GalNAc residues were substantially increased in aggressive PCa [122]. Out of at least 52 different glycan structures present on the PSA molecule, multi-sialylated LacdiNAc (GalNAcβ1-4GlcNAc, LDN) was significantly upregulated. When combined with other promising structures into a PCa-specific diagnostic model, the so-called PSA G-index, an AUC of 1.00 could be achieved for 15 PCa and 15 BPH patients, compared to 0.50 and 0.60 for tPSA and fPSA%, respectively [123]. According to a recent study performed on 204 patients (105 of whom were identified as a csPCa) by Shoji et al., not only LDN-PSA but LDN-PSAD would be a reliable biomarker for use in the range of tPSA in the grey zone (i.e., 4–10 ng mL^−1^, and also up to 20 ng mL^−1^) and for PI-RADS category 3+ [124]. The AUC of LDN-PSAD (obtained from ROC curve) has been shown to be superior to that of LDN-PSA and PSAD alone in different studies, namely 0.86 vs. 0.83 and 0.81, respectively [125] and 0.85 vs. 0.78 and 0.84, respectively [23] and could also be used successfully for distinguishing the PCa from the BPH cohort and detection of castration-resistant PCa (AUC = 0.90) [126], while the AUC of tPSA was in the range of 0.59 to 0.71. Since the pioneering study of Okada et al. in 2001, when the authors analyzed two PSA isoforms isolated from human seminal fluid (A isoform with a pI of 7.2 and B isoform with a pI of 6.9) and proved that the glycan moieties differed not only in sialic acid composition, but also in their outer glycan chain features by confirming the presence of Galβ1-4GlcNAcβ1-, GlcNAcβ1- and GalNAcβ1-4GlcNAcβ1- epitopes [127], a number of groups have been engaged in the development of a simple and user-friendly method applicable to clinical practice. Although historically PAP was involved in PCa detection prior to PSA discovery, despite the limitations of the PSA molecule in the PCa diagnostic algorithm being well known, very few studies have dealt with the glycosylation profiles of other biomarkers. One recent study by Wang et al. found that after DRE (by which urine enriched in proteins by prostate massage is obtained), PAP was analyzed by a combination of capillary electrophoresis and mass spectrometry after affinity purification from the urine and proteolytic digestion and was shown to contain three N-glycosylation sites (ASN94, ASN220 and ASN333) compared to only one in PSA (ASN69) [128].

Lectin from *Wisteria floribunda* seeds (WFL, ~116 kDa) is a tetrameric plant agglutinin shown to discriminate between cholangiocarcinoma and hepatholithiasis with a strong affinity towards terminal GalNAc residues (LacdiNAc and branched LacNAc structures) at the non-reducing terminus, with 43% sequence identity with *Bauhinia purpurea* lectin and an optimum pH at ~8.0 [129,130]. Interestingly, by preparing a recombinant (from *Escherichia coli*) WFL with its CYS272 replaced, this produced a monomer more specifically binding to LacdiNAc [131]. Its affinity towards Mac-2-binding protein (being produced by stromal cells of cirrhotic liver and correlating with hepatocellular carcinoma (HCC) via galectin-3/mTOR pathway enhancement) due to increased branching and decreased sialylation has been shown to be a promising biomarker of HCC [132,133,134]. The structural details of WFL lectins, including glycan-binding site, are depicted in Figure 5.

It should be noted that LDN has been found in normal human PSA (from seminal plasma); only the amount differs from PCa. This is also due to substrate and sugar-acceptor availability, as the synthesis competes with the LacNAc formation. Nevertheless, LDN appears to be one of the best glycan epitopes for distinguishing prostate cancer cases from benign states [123]. Several prostate cancer cell lines were shown to be able to attach LacdiNAc to their protein backbones (as confirmed by MS, Western blot and WFL-based affinity chromatography), namely 22Rv1 [19], DU145 [135], LNCaP [19,136,137], MDAPCa2b [138] and PC-3 [135]. Other benign cell lines, such as HEK293 (human embryonic kidney cells), have been shown to attach LacdiNAc moieties to their glycoproteins [139], so LacdiNAc is probably not wholly cancer-specific, but plays an unknown role in cancer biology [55]. Furthermore, except for WFL, Trichosanthes japonica agglutinin-II (TJA-II), and possibly galectin-3, also binds to β-GalNAc residues and LacdiNAc, occurring in an aggressive form of PCa [8,122]. WFL-reactive glycans as a part of clinically useful glycoproteins (including but not limited to FDA-approved oncomarkers, such as PSA) might be used in different platforms for PCa detection, such as label or label-free formats of different biosensors (e.g., two-step surface plasmon field-enhanced fluorescence spectrometry (SPFS)-based WFA lectin-anti-PSA antibody immunoassay) [140], lectin microarray [141,142] or lectin-based histochemistry [143]; the latter does not apply to liquid biopsy approaches. Since there are many different and commercially available lectins recognizing GalNAc and GlcNAc epitopes (such as Dolichos biflorus DBA and soybean agglutinin SBA, both identified as early as the late 1980s) [144,145], the need for a centralized and readily accessible reference point has been addressed in the form of the BiotechLec (https://unilectin.unige.ch/biotechlec/ (accessed on 1 September 2024)) project, which, it is hoped, will drive further progress in the area of N-acetylated sugar epitopes in situ detection (without prior release from its conjugate) to promote research and clinical development for early and accurate PCa detection [146].

Often overexpressed by cancer cells is N-acetyllactosamine (LacNAc), interacting with galectins (e.g., galectin-3 binding to Lac, LacNAc or poly-N-acetyllactosamino glycan) and contributing to carcinogenetic events, such as metastases’ formation and cancer-related adhesion or immune escape [147,148]. For Gal residues present in the LacNAc structure, there are several lectins able to bind to it (mostly those also recognizing GalNAc), even highly toxic ones such as *Ricinus communis* I lectin. Most promising, however, are two phytohaemagglutinin isoforms isolated from *Phaseolus vulgaris*, PHA-E (E = erythroagglutinin) and PHA-L (leukoagglutinin) recognizing glycans with outer Gal and bisecting GlcNAc and β(1,6)-branched tri- and tetra-antennary N-glycans, respectively [149]. PHA is a tetramer constructed from two types of peptide chains (E and L) with different preferential binding despite their structural homology, hence five different isomers with a mass of ~130 kDa exist—E4, E3L1, E2L2, E1L3 and L4 [150]. PHA-L has attracted attention due to the fact that specific N-acetylglucosaminyltransferases (GnT-III, GnT-IVs, GnT-IX (Vb) or, especially, GnT-V encoded by the MGAT5 gene, as probably the most characterized cancer-associated glycosyltransferases) play an important role in cancer progression, so this lectin can help identify a carcinogenesis early on [151,152]. In contrast, bisecting GlcNAc recognized by PHA-E, which probably inhibits the action of other GnTs, has been used for ceruloplasmin glycoprofiling in distinguishing pancreatic cancer (PC) patients from healthy individuals (AUC = 0.97) and pancreatitis patients (AUC = 0.76) [153]. However, the fact that bisecting GlcNAc structures are overexpressed in PC does not necessarily mean the same trend pertains in other cancers, although an increased expression of bisecting GlcNAc in prostatic fluids (rich in PSA and PAP) was shown during cancer progression in 2013 [154], while a decrease was observed in hormonally treated tumors [155]. Totten et al. described differences between benign hyperplasia and PCa patients in the PHA-L/E fraction of CD163, C4A and ATRN proteins, using multi-lectin affinity chromatography and quantitative mass spectrometry [156]. These findings may lead to the design and construction of novel diagnostic assays or biosensors, such as in the case of an impedimetric (electrochemical) PHA-L-based biosensor for the detection of β(1,6)-branched glycans in a cancer-related protein sample down to several μg mL^−1^ [157]. Both lectins can be found at https://www.rcsb.org/ (accessed on 1 September 2024) using pdb codes 5AVA and 1FAT. As with WFL, these lectins also bind Ca^2+^ and Mn^2+^ ions.

## 6. Conclusions

In this review, we sought to provide a general overview and rationale of the biochemical pathways of N-acetylated glycan synthesis, of which structures these saccharides form a part (due to the expression of certain enzymes during cancerogenesis); we also point out alternatives to a “heavy machinery” glycan identification using mass spectrometry, such as lectin-based methods (using WFL or PHA agglutinins). The combination of these approaches with tissue-specificity of prostate biomarkers (PSA, PSMA and PAP) is a promising and increasingly important hallmark of research in the field of glycomics.

In a recent study, we also presented evidence for economic and environmental needs related to glycomic studies on PCa. Also, in case of any future pandemic of such an extent as in the case of SARS-CoV-2, more reliable non-invasive and non-imaging methods of oncological disease detection will be needed [158]. Despite some contradictory factors mainly due to the low levels of PSA in the blood, there is a consensus on sugar chains found in PCa-derived PSA—β-N-acetylgalactosaminated (as discussed in this review), α-1,2-fucosylated and α-2,3-sialylated. Also, β-1,6-branched N-glycans are being overexpressed in PCa, as well as Tn (GalNAc-O-Ser/Thr) antigen [159]. Fucosylation was abundant in adenocarcinomas and neuroendocrine PCa eventually developing into recurrent metastatic PCa after radical prostatectomy [160]. In addition, the expression of Gal-3 is significantly decreased in tumor tissue, while Gal-4 is being overexpressed, ultimately leading to the lowered expression of E-cadherin [34,161]. Gal-4 engages with C1GALT1-(glycoprotein-N-acetylgalactosamine 3-β-galactosyltransferase 1)-dependent O-glycans to promote metastasis and castration resistance (leading to the ineffectiveness of the androgen-deprivation therapy of PCa). The interaction upregulated the expression of C1GALT1 and ST3GAL1, which in turn altered cellular O-glycome to promote the Gal-4 binding. The high expression of C1GALT1 and Gal-4 is thus a prediction of poor survivability [162]. Another enzyme correlating with tumor growth is GALNT7, which has been shown to be an important driver of disease progression [162].

The above-mentioned glycan epitopes are a promising tool in diagnosing PCa—especially in a combination, as shown in our clinical studies [163,164]. Some specific assay configurations can be applied to overcome common analytical barriers, such as low analyte concentrations or non-specific interactions, namely, the implementation of magnetic nano-/microparticles [165]. The implementation of N-acetylated monosaccharide-containing glycan epitopes into a diagnostic algorithm of early and metastatic PCa (using PHA and WFL lectins) is a promising area for future clinical research for various secreted and membrane-bound biomarkers.

## Figures and Tables

**Figure 1 cancers-16-03786-f001:**
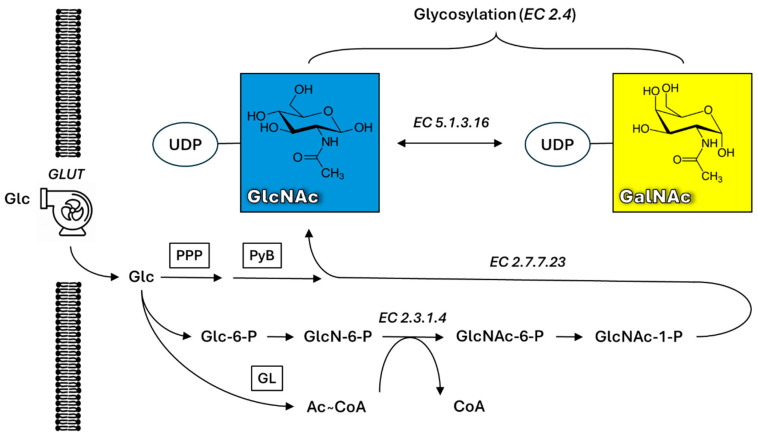
Schematic presentation of glucose metabolism and its linkage to glycosylation involving GlcNAc and GalNAc structures from activated UDP-saccharides. Glc = glucose; GlcN = glucosamine; GlcNAc = N-acetylglucosamine; GalNAc = N-acetylgalactosamine; GLUT = glucose transporter; Ac = acetyl group; PPP = pentose phosphate pathway; PyB = pyrimidine biosynthesis; GL = glycolysis; EC 2.3.1.4 = GlcN-6-P N-acetyltransferase; EC 2.7.7.23 = UDP-GlcNAc pyrophosphorylase; EC 2.4 = glycosyltransferases. To ref. [18].

**Figure 2 cancers-16-03786-f002:**
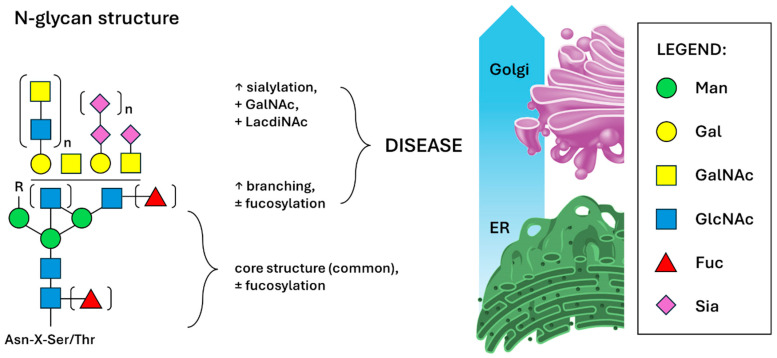
Schematic presentation of N-glycan biosynthesis in vivo. A common core structure containing trimannosyl core is further decorated with a sequence of glycan-modifying enzymes in Golgi, which can be differently expressed during disease progress. R = different glycan antennae based on glycan type (i.e., high-mannose, complex and hybrid type); X = amino acid other than PRO; ER = endoplasmic reticulum; Man = mannose; Gal = galactose; GalNAc = N-acetylgalactosamine; GlcNAc = N-acetylglucosamine; Fuc = fucose; Sia = sialic (N-acetylneuraminic) acid. The figure is based on refs. [11,54,55,56].

**Figure 4 cancers-16-03786-f004:**
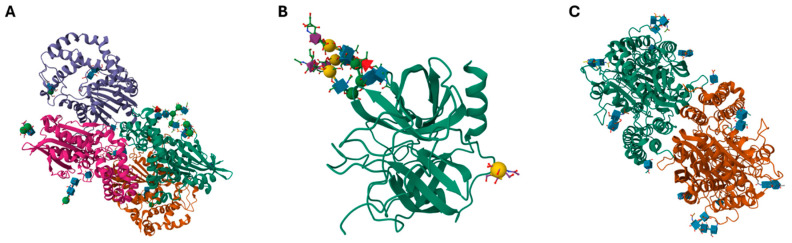
Presentation of three well-known PCa biomarkers, namely extracellular (**A**) PAP (pdb code 1CVI, EC 3.1.3.2, 167.2 kDa) and (**B**) fPSA (pdb code 3QUM, EC 3.4.21.77, 30–34 kDa) proteins and membrane-bound (**C**) PSMA (pdb code 1Z8L, EC: 3.4.17.21, 324.15 kDa). Source: https://www.rcsb.org/ (Protein Data Bank, accessed on 1 September 2024). Glycans attached to protein backbone, recognizable by lectins, are clearly visible.

**Figure 5 cancers-16-03786-f005:**
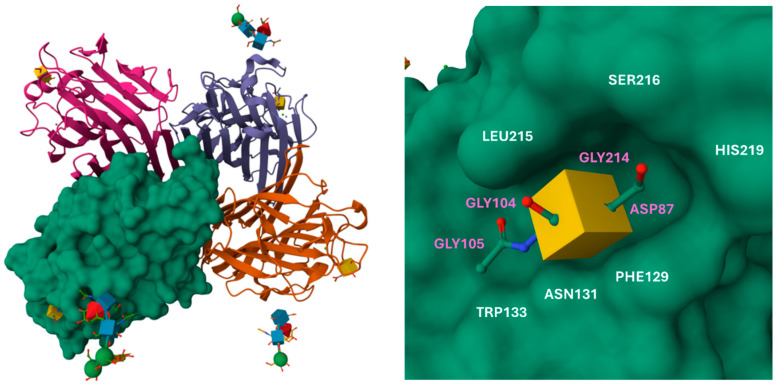
(**Left**) presentation of *Wisteria floribunda* lectin (glycoprotein, pdb code 5KXB) and its tetrameric structure with one monomeric chain highlighted as a molecular surface (green). (**Right**) details of GalNAc-binding site with depicted amino acid residues involved in the saccharide binding (pink residues are involved in the binding directly in the cavity, beneath the GalNAc structure). Source: https://www.rcsb.org/ (Protein Data Bank, accessed on 1 September 2024). The structure also binds Ca^2+^ and Mn^2+^ metal ions.

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
