# Peer review of "N-Acetylated Monosaccharides and Derived Glycan Structures Occurring in N- and O-Glycans During Prostate Cancer Development"

_cancers, 2024, doi:10.3390/cancers16223786_

Round 1
Reviewer 1 Report
Comments and Suggestions for Authors
In the manuscript "N-acetylated monosaccharides and derived glycan structures present on serum oncomarkers during prostate cancer development," the authors provide an overview of abnormal glycosylation's role in PCa screening and prognosis. While the manuscript is well-written overall, I have several important recommendations for improvement:
- A significant portion of the manuscript focuses on atypical glycans within the prostate tumor itself. Therefore, I suggest revising the title to encompass not only serum glycan epitopes but also both N- and O-glycosylation.
- The role of Golgi dispersal and ER stress in the formation of these glycans is well-established but described too superficially here. A deeper discussion is needed.
- The contribution of MGAT5-induced glycan formation could be addressed more rigorously, incorporating recent PCa-relevant studies.
- The manuscript would benefit from a table describing the specific glycan epitopes with corresponding references to aid reader comprehension. Additionally, figures illustrating pathways (e.g., MGAT3, MGAT5, B4GalNTs) would enhance clarity. Figure 2, however, adds little value and could be omitted, as the publicly available structures of proteins or lectins don’t seem to provide critical information.
- The role of glycans and their specific features in metastatic PCa is missing, though it is an area with notable insights. This should be included.
- I recommend adding a graph that tracks the evolution of glycan analysis over the years, with emphasis on clinical implications.
- Controversial results in PCa glycan analysis, such as PSA glycan analysis used for diagnosis, are not addressed and should be discussed.
- The authors should also highlight their own contributions to this field. Why not reference some of their published work to offer specific insights? Consider asking: What new knowledge are we contributing to this field with this review article?
N/A
Author Response
Please see the response to the Reviewer's comments in the attached file.

Reviewer 2 Report
Comments and Suggestions for Authors
In this article, the authors review the potential of glycosylation for prostate cancer biomarker discovery.
General comment:
The topic of this review paper is highly relevant. The manuscript is well written with a lot of information presented and it is a valuable source of information (especially section 5) on the potential of glycosylation for prostate cancer biomarker discovery.
Minor points:
Lines 190-191: The authors could mention the importance of the EMT for cancer biology.
Line 192: It would be good to mention the origin of tissue/cell line in which 'decreased sialylation, particularly on β4-integrins' was observed upon 'TGF-β-induced EMT'. Was this a cancer tissue/cell line? Does this decreased sialylation have a functional consequence? What is known about EMT and glycosylation in prostate cancer? Just like for EMT, maybe it would be good to add a bit more (several sentences) on metastasis process and the role of glycosylation in it, particulary with the data on prostate cancer.
Lines 196-198: The same comment as above; it would be good to mention the origin of a tissue/cell line.
Author Response

(The authors gave the same response as above.)

Round 2
Reviewer 1 Report
Comments and Suggestions for Authors
The authors improved the manuscript.